# The smallest known Devonian tetrapod shows unexpectedly derived features

## Per E. Ahlberg[1] and Jennifer A. Clack[2,†]

[1]Department of Organismal Biology, Uppsala University, Norbyvägen 18A, 752 36 Uppsala, Sweden
[2]University Museum of Zoology Cambridge, Downing Street, Cambridge CB2 3EJ, UK

 PEA, 0000-0001-9054-2900; JAC, 0000-0003-0017-5831

palaeontology

vertebrate, tetrapod, jaw, Devonian, Greenland

**Author for correspondence:**
Per E. Ahlberg
e-mail: per.ahlberg@ebc.uu.se

A new genus and species of Devonian tetrapod, *Brittagnathus minutus* gen. et sp. nov., is described from a single complete right lower jaw ramus recovered from the *Acanthostega* mass-death deposit in the upper part of the Britta Dal Formation (upper Famennian) of Stensiö Bjerg, Gauss Peninsula, East Greenland. Visualization by propagation phase contrast synchrotron microtomography allows a complete digital dissection of the specimen. With a total jaw ramus length of 44.8 mm, *Brittagnathus* is by far the smallest Devonian tetrapod described to date. It differs from all previously known Devonian tetrapods in having only a fang pair without a tooth row on the anterior coronoid and a large posterior process on the posterior coronoid. The presence of an incipient surangular crest and a concave prearticular margin to the adductor fossa together cause the fossa to face somewhat mesially, reminiscent of the condition in Carboniferous tetrapods. A phylogenetic analysis places *Brittagnathus* crownward to other Devonian tetrapods, adjacent to the Tournaisian genus *Pederpes*. Together with other recent discoveries, it suggests that diversification of 'Carboniferous-grade' tetrapods had already begun before the end of the Devonian and that the group was not greatly affected by the end-Devonian mass extinction.

[†]Deceased 26 March 2020.

## 1. Introduction

In 1987, an expedition to East Greenland mounted by the University of Cambridge and the Geological Museum Copenhagen, under the auspices of the Greenland Geological Survey, collected a large number of specimens of the Devonian tetrapods *Acanthostega* and *Ichthyostega* [1]. These have been described in much detail in a series of publications [2–13]. A paper by the current authors and

**Figure 1.** (*a*) Map of Greenland showing locality where NHMD 116368 was collected (red asterisk). Modified from [15]. (*b*) Photograph of specimen in internal and external view. (*c–e*) Images from scan dataset. (*c*) Series of transverse sections of the jaw, arranged from anterior (left) to posterior (right). In each section, the lateral surface of the jaw is to the right. (*d*) Longitudinal section through dentary, showing teeth. (*e*) Longitudinal section through entire jaw, showing articular and Meckelian fenestrae. (*d*) and (*e*) are shown to the same scale. Scale bars in (*b*) and (*e*), 10 mm.

colleagues [14] described and named a third taxon of tetrapod, *Ymeria denticulata*, distinguished principally by its lower jaw anatomy. Here, we describe a fourth taxon, represented by a single lower jaw ramus.

This specimen was collected during the 1987 expedition, but because of its small size, mechanical preparation was halted for the fear of damage to the delicate dentition. Investigation by propagation phase contrast synchrotron microtomography (PPC-SRµCT) has now made anatomy accessible for the first time and thus allowed description and comparison with other taxa.

The specimen is significant in a number of ways. Firstly, as the fourth tetrapod taxon to be discovered in the Late Devonian (Famennian) strata of East Greenland, it makes this the most taxonomically diverse Devonian tetrapod assemblage known. Secondly, it derives from the Britta Dal Formation, from the same mass-death deposit that has yielded most of the *Acanthostega* material [15]. It thus changes our perception of this deposit, which has previously appeared to be completely monospecific. Thirdly, it shows a number of derived characters more in keeping with those of early Carboniferous (Mississippian) forms, echoing similar findings among lungfish taxa [16], and the reverse in early Carboniferous forms such as *Lethiscus* and *Perittodus* that show plesiomorphic characters more typical of Late Devonian forms [17,18]. Finally, its small size, in what is obviously an adult animal to judge by its state of ossification, is unprecedented in Devonian tetrapods so far known.

## 2. Material and methods

The specimen, NHMD 116368, (field number MGUH f.n. 1373) was collected in July 1987 from the *Acanthostega*-yielding horizon on Stensiö Bjerg (figure 1*a*). Originally measured at 770 m altitude, a

**Table 1.** Abbreviations

| | |
|---|---|
| ad.fos | adductor fossa; |
| adsym | adsymphysial plate; |
| adsym.fa | fang pair of adsymphysial plate; |
| ang | angular; |
| art | articular; |
| cor1 | anterior coronoid; |
| cor1.fa | fang pair of anterior coronoid; |
| cor2 | middle coronoid; |
| cor2.fa | fang pair of middle coronoid; |
| cor3 | posterior coronoid; |
| cor3.pp | posterior process of posterior coronoid; |
| cor3.te | teeth of posterior coronoid; |
| de | dentary; |
| de.fa | dentary fang; |
| de.te | dentary teeth; |
| lat.line | lateral line groove; |
| lat.pore | lateral line pore; |
| Meck | Meckelian bone; |
| Meck.fen | Meckelian fenestrae; |
| pospl | postsplenial; |
| preart | prearticular; |
| spl | splenial; |
| surang | surangular; |
| surang.cr | surangular crest; |
| sym | symphysial exposure of Meckelian bone |

subsequent expedition in 1998 recalibrated this to 800 m. The matrix is the typical micaceous silty sandstone of that horizon. It was initially prepared mechanically, which revealed the posterior end of the lateral surface and the lower margin and prearticular on the medial surface (figure 1b).

In September 2016, the specimen was imaged using PPC-SR$\mu$CT at the European Synchrotron Radiation Facility (ESRF) in Grenoble, France. The scan was part of Experiment ES-505 and was made courtesy of Valéria Vaškaninová to whom the beam time had been awarded. The voxel size of the scan was 13.49 μm, made with single distance phase retrieval and a propagation distance of 5 m. The reconstructed volume was converted into a stack of 16 bit TIFF images (figure 1c; abbreviations are given in table 1). Segmentation was performed using Mimics Research 19.0 (Materialise Software); STL files of the segmented structures were exported to Blender 2.79b for rendering.

The phylogenetic analysis was performed using a data matrix modified from that in [19] (see electronic supplementary material, Appendices 1 and 2; NEXUS File). The matrix contains 109 characters scored for 29 taxa (the new taxon plus the taxa used by [19]). *Aytonerpeton*, *Diploradus* and *Perittodus*, which were not present in the matrix of [19], were coded from [18]. The recently described *Parmastega* was coded from [20]. The analyses were performed using PAUP* 4.0b [21], with characters 32, 50, 65, 67, 73, 82, 95, 106 and 107 coded as ordered and *Eusthenopteron* specified as the outgroup. A reduced matrix omitting *Aytonerpeton*, *Diploradus* and *Perittodus* was analysed using a branch-and-bound algorithm; the full matrix with these genera included proved unmanageable for the computer using branch-and-bound, and was instead subjected to a heuristic search. The heuristic settings were as follows: optimality criterion = parsimony; starting tree(s) obtained via stepwise addition; addition sequence simple (reference taxon = *Acanthostega*); number of trees held at each step during stepwise addition = 1; branch-swapping algorithm: tree-bisection-reconnection; steepest descent option not in effect; topological constraints not enforced; trees are unrooted. Tree support was estimated by the bootstrap method, using a heuristic search (with settings as above) of 1000 bootstrap replicates.

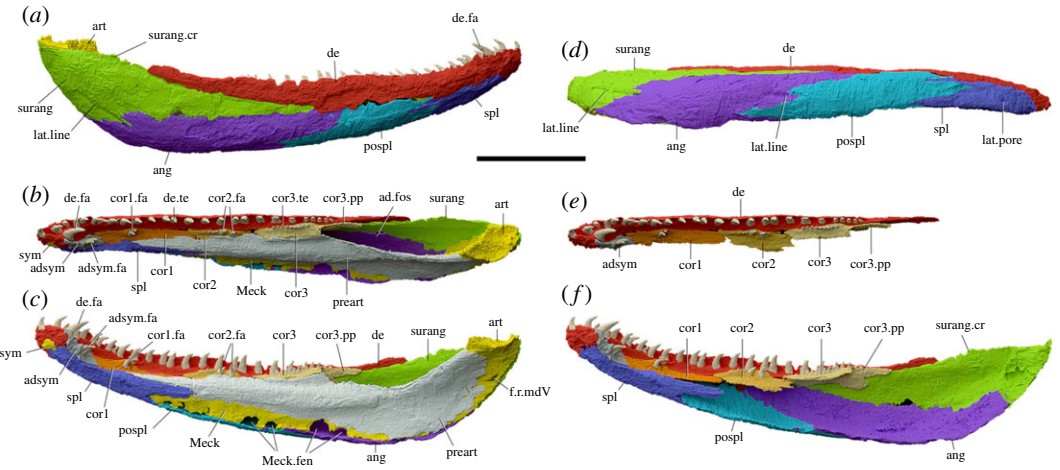

**Figure 2.** Digital reconstruction of the jaw, subdivided into separate bones, segmented in Mimics Research 19.0 and rendered in Blender 2.79b. (*a*) Lateral view. (*b*) Dorsal view. (*c*) Medial view. (*d*) Ventral view. (*e*) Dorsal view without prearticular, Meckelian bone and infradentaries. (*f*) Medial view without prearticular and Meckelian bone. Scale bar, 10 mm.

# 3. Results

## 3.1. Systematic palaeontology

Osteichthyes Huxley, 1880

 Sarcopterygii Romer, 1955

 Tetrapodomorpha Ahlberg, 1991

 Tetrapoda Jaekel 1909

 Genus *Brittagnathus* gen. nov.

 *Genus*: LSID registration number urn:lsid:zoobank.org:act:ADE45E01-38FE-49A2-B65D-ED3BBA759E0E.

 *Etymology*: from the Britta Dal Formation of the Upper Devonian of East Greenland, where the specimen was collected, and New Latin 'gnathus' (originally Greek 'gnathos'), jaw. The formation is named after Britta Säve-Söderbergh (née Arnell), wife of Gunnar Säve-Söderbergh.

 *Diagnosis*: defining unique combination of characters includes 43 marginal tooth positions plus symphysial fang pair on dentary; adsymphysial and anterior coronoid carry fang pairs but no tooth row; small surangular crest; prearticular with well-developed horizontal ridge; prearticular margin of adductor fossa concave in mesial view.

 Characters shared with *Greererpeton* and more crownward post-Devonian tetrapods: adductor fossa faces mesially, margin concave in mesial view; abruptly tapering or 'stepped' dentary ventral edge; no tooth row on adsymphysial plate; prearticular shagreen scattered patches or absent.

 Plesiomorphies shared with Devonian fishes or tetrapods: Meckelian element continuous but developed into distinct symphysial and articular ossifications; middle coronoid carries fang pair set into short tooth row; prearticular contacts angular edge to edge without suture. Prearticular-surangular contact absent.

 *Type and only known species*: *Brittagnathus minutus* sp. nov.

 *Species*: LSID registration number urn:lsid:zoobank.org:act:59701DB3-A3EA-48CF-8E76-F05DA78180C9

 *Etymology*: from Latin 'minutus', tiny, reflecting the small size of the specimen.

 *Diagnosis*: as for the genus, by reason of monotypy.

 *Holotype and only specimen*: Natural History Museum, Denmark (NHMD) 116368. A small complete right lower jaw ramus.

 *Type locality and horizon:* Britta Dal Formation, Gauss Peninsula, East Greenland, at 800 m on the southeast side of Stensiö Bjerg. Late Devonian, late Famennian.

## 3.2. Description

The jaw is 44.8 mm long (figures 1*b* and 2). It is robustly built, and only the suture between the dentary and the infradentary series can be seen on the surface. The sutures between the infradentaries are only visible in the scan and are strongly interdigitated. The lower margin curves medially, and the overall

shape of the jaw appears natural, although the deep overlap of the prearticular onto the middle coronoid and the somewhat displaced position of the latter bone suggests that the middle part of the jaw may have been slightly compressed labiolingually (figure 2b,c,e,f). External ornament is poorly developed except at the symphysial end where pits and ridges can be made out. Although much of the lateral surface was exposed at the time of collection, it does not seem to be eroded or worn.

The lateral surface is of typical early tetrapod construction with four infradentaries (figure 2a,d). The surangular is long, about half the length of the jaw ramus. It rises slightly posteriorly, to form what could be considered an incipient surangular crest. The angular reaches a little further forward beneath it. The postsplenial is narrow and relatively short and the splenial is splint-like. Because the infradentaries are lightly ossified and affected by internal crystal growth (figure 1c), which creates a 'speckled' effect that obscures anatomical structures, it is very difficult to trace the course of the mandibular lateral line. However, short lateral line grooves are visible at the surangular–angular and angular–postsplenial sutures, and there appears to be a single lateral line pore on the splenial (figure 2a,d).

The dentary is narrow posteriorly but grows broader and deeper anteriorly; at its mid-point, there is an abrupt increase in depth. Its tooth row comprises 43 tooth positions, of which 26 are occupied by fully grown teeth, one by a developing tooth, and 16 by empty sockets (figure 2b,c). There is a dentary fang and associated replacement pit internal to the main tooth row. The inner dental arcade consists of an adsymphysial plate and three coronoids (figure 2b,c,e,f). The adsymphysial plate and anterior coronoid each carries only a small fang pair, whereas the middle coronoid has a short tooth row with a fang pair at the posterior end. The posterior coronoid carries a tooth row without fangs. When fangs are present, the bone is developed into a thin vertical crest that carries the fang pair on its mesial flank. The main part of each coronoid is a flat and rather narrow base plate; the posterior coronoid meets the prearticular edge to edge, but the middle and anterior coronoids are overlapped by the prearticular to varying degrees and thus appear even narrower in dorsal view than they actually are. The posterior coronoid forms a bifid posterolateral process that extends along the surangular on the lateral surface of the adductor fossa, but is not exposed on the lateral surface of the jaw. The middle coronoid is somewhat displaced ventrally, suggesting a small amount of collapse and lateral compression.

The mesial surface (figure 2c) is dominated by the prearticular, which bears a smooth-surfaced thick and prominent longitudinal ridge that occupies approximately the anterior two-thirds of the length of the bone. Dorsal to this ridge the bone thins into a flange that contacts the posterior coronoid and overlaps the middle and anterior coronoids. The anterior end of the prearticular is level with the fang pair on the anterior coronoid; the posterior end does not quite reach the posterior margin of the articular and is thus well separated from the surangular. Posteroventrally the prearticular contacts the angular, but does not suture with it. No denticles are present on the prearticular or anywhere else on the jaw.

The adductor fossa is relatively long, occupying approximately one third of the length of the jaw (figure 2b,c). Its lateral wall is formed principally by the surangular and angular. The dentary extends along approximately 60% of the length of the fossa, but is slender and almost excluded from the fossa itself by the surangular, which lies mesial to it. A small component of the adductor fossa wall is formed by the posterior process of the posterior coronoid, which overlaps the surangular and dentary.

Much of its lateral side is flanked by the posterior end of the dentary.

The dorsal margin of the prearticular that forms the mesial edge of the adductor fossa dips to a low point just posterior to the end of the longitudinal ridge, then rises again as it passes anteriorly towards the contact with the posterior coronoid. Together with the slight surangular crest, this imparts a dorsomesial orientation to the fossa.

The Meckelian bone is ossified in its entirety, allowing all of it to be reconstructed from the scan data, but the ossification is much stronger at the articular and symphysial ends than in the middle part. The articular, which is especially well ossified and carries a well-defined bilobed glenoid, is exposed between the prearticular and the posterior margin of the jaw, and projects slightly above the dorsal jaw margin. The Meckelian bone exposure in the middle part of the jaw is relatively high, reflecting the gently arched ventral margin of the prearticular, and contains several foramina of varying sizes which are bounded ventrally by the postsplenial and angular. The splenial bounds the Meckelian exposure anteriorly and sends a posterodorsomesial process between this and the prearticular. It does not contact the middle coronoid. Anteriorly the Meckelian bone of the symphysis is exposed.

## 4. Comparison with Devonian and Carboniferous tetrapods

The first comparative point that needs to be established is the distinctness of *Brittagnathus* from the previously known Greenland tetrapods *Acanthostega*, *Ichthyostega* and *Ymeria*. *Acanthostega* is of

particular interest here, because it completely dominates the deposit where the *Brittagnathus* jaw was found; it is thus important to rule out the hypothesis that *Brittagnathus* is simply a juvenile *Acanthostega*. Fortunately, the lower jaw of *Brittagnathus* differs from that of *Acanthostega* [12,13] in a number of respects beyond its small size and the apparently greater degree of Meckelian ossification. The ramus is proportionately more robust in *Brittagnathus* than in *Acanthostega* and the region of the adductor fossa is deeper. Partly as a consequence of this, there is a sharper change in the curvature of the ventral jaw margin at the posterior end of the angular, creating a visible transition from a ventrally facing to a posteroventrally facing segment of the margin. In *Acanthostega*, the same region of the jaw margin forms a smooth curve without an obvious inflection point. In *Brittagnathus*, the posterior end of the angular is located some distance posterior to the end of the dentary, approximately level with the anterior margin of the glenoid; in *Acanthostega*, by contrast, the dentary extends further posteriorly than the angular and the angular does not reach the level of the glenoid.

Further differences can be seen in mesial view. Below the adductor fossa, the prearticular of *Brittagnathus* reaches down to achieve an edge contact (but not a suture) with the angular, concealing the Meckelian bone from view. This condition is also seen in some other Devonian and Carboniferous tetrapods including *Parmastega*, *Ichthyostega*, *Ventastega* and *Whatcheeria* [12,20,22], while in *Densignathus* the prearticular extends ventrally but nevertheless fails to reach the angular [23]. In *Acanthostega*, however, the ventral expansion of the prearticular is very weakly developed and is separated from the edge of the angular by a wide gap that was presumably filled with Meckelian cartilage in life. Three other differences in the region of the adductor fossa are also noteworthy. Firstly, the fossa itself is proportionately larger in *Brittagnathus* than in *Acanthostega*; secondly, the prearticular margin of the fossa is anteroposteriorly concave, whereas in *Acanthostega* (and all other Devonian tetrapods except, arguably, *Parmastega* [12,20]), it is straight; and thirdly, the dorsal margin of the surangular is developed into a small surangular crest, whereas in *Acanthostega* and other Devonian tetrapods, it is straight. The overall effect of the last two features is that the opening of the adductor fossa faces somewhat mesially in *Brittagnathus*, but purely dorsally in *Acanthostega*. In this regard, *Brittagnathus* differs from all previously known Devonian tetrapods but resembles a range of early Carboniferous forms [12,19]. This transformation may relate to the beginning of the differentiation of the jaw adductor musculature towards the crown tetrapod condition [12].

Although the complement of tooth-bearing bones (dentary, adsymphysial plate and three coronoids) is the same in *Brittagnathus* as in *Acanthostega*, there are marked differences in the dentition. The dentary of *Brittagnathus* has only 43 marginal tooth positions, as against 59 in *Acanthostega* [13]. The anterior and middle coronoids of *Brittagnathus* carry distinct fang pairs, which are not present in *Acanthostega*. Conversely, *Acanthostega* has a continuous tooth row that extends from the fang pair on the adsymphysial plate to the posterior end of the posterior coronoid, whereas in *Brittagnathus* there is no tooth row on the adsymphysial plate or anterior coronoid, and the middle coronoid carries only a short tooth row centred on the fang pair. In summary, even though the single jaw ramus of *Brittagnathus* was discovered at a locality overwhelmingly dominated by *Acanthostega*, the hypothesis that it represents a juvenile individual of *Acanthostega* can be rejected.

Comparisons of *Brittagnathus* with *Ichthyostega* [12,24] and *Ymeria* [14] also reveal substantial differences. *Ichthyostega* has approximately 29–32 marginal dentary teeth, the adsymphysial plate carries three teeth, and the coronoids all carry tooth rows but no fangs. Furthermore, the morphology of the coronoids themselves is quite different from those of *Brittagnathus*. In *Ichthyostega*, they are near-vertical plates, separated by oblique anterodorsal–posteroventral sutures, with the tooth rows carried on the dorsal margins; in *Brittagnathus*, the coronoids are essentially horizontal plates and the sutures between them are transverse rather than oblique. *Ymeria* has a dentary tooth count of approximately 32–33 and a continuous tooth row (containing small fang pairs) extending across all three coronoids [14]. The adsymphysial plate carries a fang pair plus a smaller tooth, and the prearticular has a well-developed denticle band. The status of *Brittagnathus* as a new genus, distinct from the previously known Famennian tetrapods from Greenland, is thus robustly supported.

It has already been noted above that the slight surangular crest and distinctly concave prearticular margin of the adductor fossa distinguish *Brittagnathus* from all previously described Devonian tetrapods. The same holds true for the lack of a tooth row on the anterior coronoid and the presence of a posterior process on the posterior coronoid. By contrast, all four features are seen quite commonly in Carboniferous tetrapods, though not always together. In *Aytonerpeton* from the Tournaisian of Britain, the dentition of the adsymphysial plate and anterior coronoid is almost identical to that of *Brittagnathus*, consisting of a small fang pair on each bone [18] (note that the fang pair on the anterior coronoid is mis-labelled as 'Cor 2 tooth' in fig. 4d of [18]). The adductor fossa of *Aytonerpeton* is unfortunately

unknown. In *Whatcheeria* [22], the anterior coronoid carries a tooth row, the posterior coronoid lacks a posterior process, and the surangular lacks a crest, but the prearticular margin of the adductor fossa is gently concave. Baphetids have adductor fossae combining a concave prearticular margin and a modest surangular crest, and their coronoids are completely edentulous [12]. In *Gephyrostegus*, the adductor fossa resembles an exaggerated version of the *Brittagnathus* condition, with a much larger surangular crest and more emphatically concave prearticular margin, but with a posterior process on the posterior coronoid similar to that of *Brittagnathus* [12]. All three coronoids carry denticle shagreen, the anterior and middle coronoids also fang pairs. An adsymphysial plate is absent.

As can be seen, the lower jaw of *Brittagnathus* shows a general similarity to a number of different Carboniferous tetrapods without precisely matching any one of them. Above, we have focused on the distribution of particular characters that appear to be derived relative to the condition in 'typical' Devonian tetrapods, but the same point applies more broadly. For example, the Tournaisian genus *Perittodus* [18] has a middle coronoid dentition very similar to that of *Brittagnathus*, combining a fang pair with a short tooth row on a raised crest, but unlike *Brittagnathus* it appears also to have a tooth row on the anterior coronoid. The one respect in which *Brittagnathus* differs strikingly from the majority of Carboniferous and later tetrapods, and instead aligns with Devonian taxa, is the absence of a mesial lamina of the angular. Such a lamina, which wraps around the ventral jaw margin and sutures with the prearticular on the mesial face of the jaw ramus, is present—where this region is known—in almost all post-Devonian tetrapods except *Whatcheeria* and possibly *Diploradus* [12,18,19,22,25].

## 5. Phylogenetic analysis

As discussed in Material and methods, two phylogenetic analyses were performed: a 26-taxon data matrix was analysed using a branch-and-bound algorithm, while a larger 29-taxon matrix (also incorporating data from *Aytonerpeton*, *Diploradus* and *Perittodus*) was subjected to a heuristic analysis (figure 3). The branch-and-bound analysis of the 26-taxon matrix generated 72 trees of 271 steps. Both strict consensus (figure 3a) and 50% majority rule consensus (not shown) trees placed *Brittagnathus* in an unresolved polychotomy with *Pederpes*, immediately above *Whatcheeria* and below *Sigournea*. The more crownward tetrapods were well resolved, with only a single trichotomy of *Sigournea*, *Greererpeton* and higher tetrapods. Stemward to *Whatcheeria*, the Devonian tetrapods were fully resolved apart from a clade comprising *Densignathus*, *Elginerpeton*, *Ichthyostega*, *Metaxygnathus* and *Ymeria*, which was internally unresolved. *Parmastega* was placed as the sister group to all other tetrapods.

The 29-taxon matrix yielded 130 trees of 283 steps (figure 3b). The position of *Brittagnathus* was unchanged compared to the smaller matrix, and the overall structure of the tree remained similar, but the addition of the Tournaisian taxa caused a number of minor topological changes. *Diploradus* was positioned as sister taxon to *Crassigyrinus*, *Aytonerpeton* in a trichotomy with *Greererpeton* and more crownward tetrapods, and *Perittodus* in a stemward position among the Devonian tetrapods.

A bootstrap analysis of 1000 bootstrap replicates (see Material and methods) of the 26-taxon matrix failed to resolve the Devonian tetrapods plus 'whatcheeriids' (*Whatcheeria*, *Pederpes* and *Brittagnathus*), and also failed to recover the branch segment separating *Crassigyrinus* + *Tantallognathus* from *Sigournea* + *Greererpeton*, but all other nodes from the strict consensus tree were supported with bootstrap values varying between 51 and 95 (figure 3c). The weak support for the middle region of the tree probably reflects a combination of extensive homoplasy and the extreme incompleteness of some of these taxa. Nevertheless, we see no reason to reject the position of *Brittagnathus* recovered by the strict consensus tree.

## 6. Discussion

The discovery of *Brittagnathus* throws new light on both the East Greenland ecosystem and the development of tetrapod faunas across the Devonian–Carboniferous boundary. The late Famennian strata of East Greenland can now be seen to contain at least four tetrapods (*Ichthyostega*, *Acanthostega*, *Ymeria* and *Brittagnathus*). Of these, *Ichthyostega* is known to be contemporary with *Acanthostega* (the two have substantially overlapping ranges and have been found together at one locality in the Aina Dal Formation [15]), and *Acanthostega* with *Brittagnathus*; the single specimen of *Ymeria* is poorly constrained stratigraphically, as it was found on a talus [14], but is almost certainly contemporary with *Ichthyostega* and *Acanthostega*. In other words, all four taxa can be regarded as broadly contemporary inhabitants of the same landscape. This makes the East Greenland assemblage the most

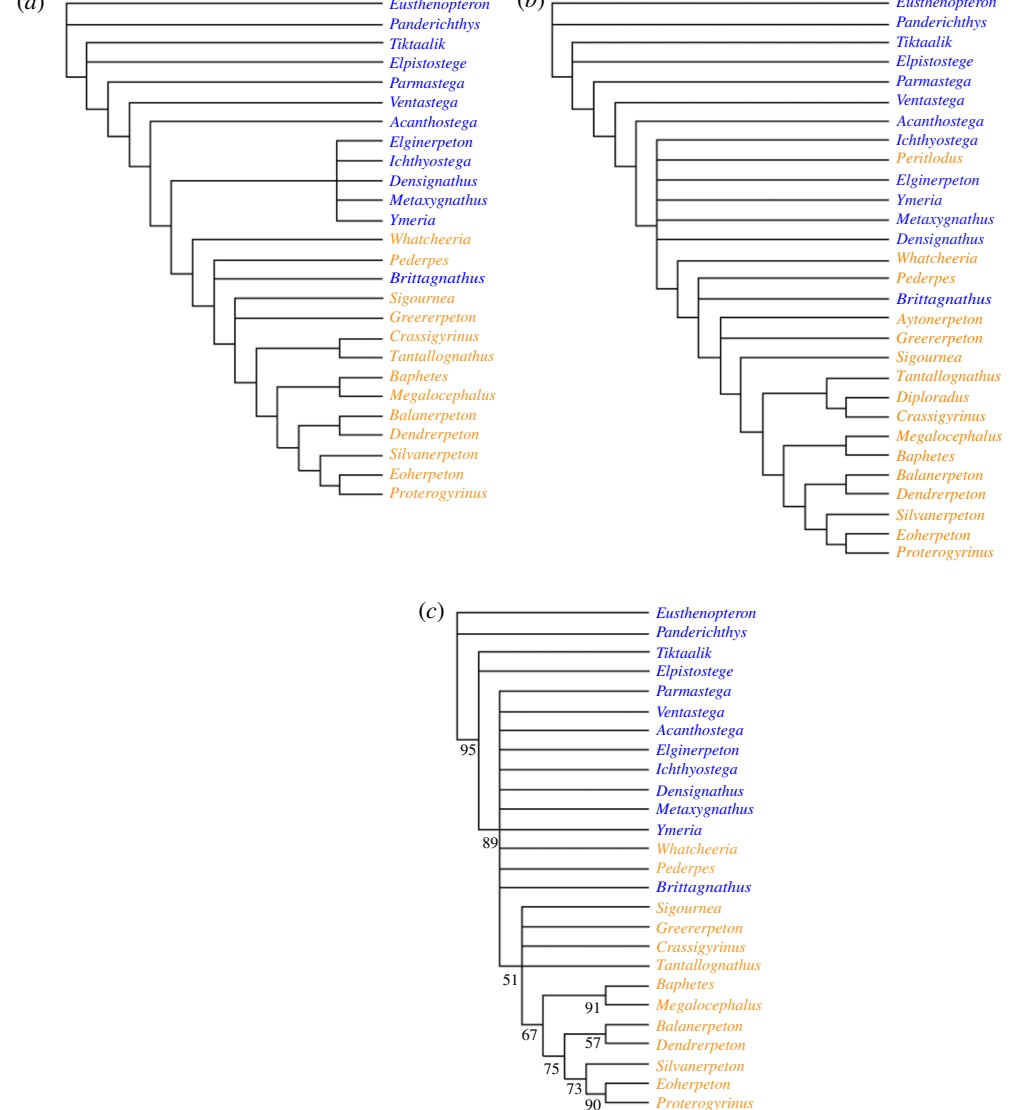

**Figure 3.** Phylogenetic analysis. (a) Strict consensus of 72 trees of 271 steps, branch-and-bound analysis of 26-taxon matrix. Consistency Index 0.491, retention Index 0.700, rescaled consistency index 0.344, Homoplasy Index 0.509. (b) Strict consensus of 130 trees of 283 steps, heuristic analysis (see Material and methods for analytical parameters), 29-taxon matrix. Consistency index 0.470, retention index 0.688, rescaled consistency index 0.323, homoplasy index 0.530. (c) Bootstrap analysis of 26-taxon matrix, 1000 replicates (see Material and methods for analytical parameters). Numbers at nodes indicate bootstrap support values. In all trees, Devonian taxa are shown in blue, Carboniferous taxa in orange.

diverse Devonian tetrapod assemblage currently known, at least in terms of described taxa, though the more fragmentary tetrapod material from the contemporary Catskill Formation in Pennsylvania hints at similar levels of diversity [26–29].

*Brittagnathus* was collected from the same locality as the famous *Acanthostega* mass-death deposit, which contains at least several tens of *Acanthostega* individuals (and possibly much more, as the site has not been fully excavated), concentrated in what appears to be a small channel deposit [15]. *Acanthostega* seems to have been largely or entirely aquatic; the deposit is interpreted as representing a school of individuals, probably sub-adults [30], which became trapped in the channel by falling water levels following a flood event. However, it is doubtful whether the *Brittagnathus* individual arrived during the same event. The fact that only a single lower jaw ramus has been found, contrasting with the articulated to semi-articulated condition of most of the *Acanthostega* material, suggests that the *Brittagnathus* jaw may belong to the 'background' of less complete fossils already scattered through the sediment prior to the arrival of the *Acanthostega* school. It is thus not possible to draw any detailed conclusions about the environmental preferences of *Brittagnathus*, or its ecological relationship (if any) to *Acanthostega*.

Turning to the evidence provided by the lower jaw ramus itself, it is obvious from the dentition that *Brittagnathus* was a predator. Its dentary teeth are proportionally more robust and widely spaced (hence less numerous) than those of *Acanthostega*, and also different in shape, being recurved rather than upright. This suggests subtly different feeding preferences, perhaps with a focus on proportionally larger prey. The mandibular lateral line canal is difficult to visualize in the scan, but it is certainly present and appears to consist of short open grooves crossing the infradentary sutures, alternating with enclosed sections of canal in the centres of these bones. *Brittagnathus* was thus at least partly aquatic [31].

The most noteworthy attribute of the jaw is its small size. If we assume similar body proportions to *Ichthyostega* and *Acanthostega* [32], the total body length of *Brittagnathus* was approximately 25 cm. This makes it by far the smallest Devonian tetrapod described to date. However, already in the early Carboniferous, small tetrapods were commonplace. Among the Tournaisian tetrapods from the Ballagan Formation of the Scottish Borders, *Diploradus* and *Aytonerpeton* are similar in size to *Brittagnathus*, whereas *Koilops* and *Perittodus* are approximately 50–100% larger in linear dimensions but still much smaller than typical Devonian genera like *Ichthyostega* and *Acanthostega* [18]. The Viséan aïstopod *Lethiscus* has a lower jaw slightly shorter than that of *Brittagnathus* [17], and of course, the later Viséan tetrapod assemblage from East Kirkton is completely dominated by small forms such as *Balanerpeton*, *Silvanerpeton* and *Westlothiana* [33–37].

*Brittagnathus* is not the only 'whatcheeriid-grade' tetrapod known from the Famennian. A postorbital and a jugal from the Catskill Formation of Pennsylvania resemble those of *Pederpes*, while a femur from the same formation resembles that of *Ossinodus* [27]. A maxilla from Becco in Belgium has also been described as 'whatcheeriid-like' [38]. *Tulerpeton* from the terminal Famennian of Andreyevka-2 in Russia is clearly more crownward than all other described Devonian tetrapod taxa [39,40] and can perhaps be placed in the 'whatcheeriid' grade. However, all these animals are larger than *Brittagnathus*. This is a point of considerable interest, because it has long been noted that Carboniferous tetrapod assemblages differ from Devonian ones both in the more derived morphology of the animals and in their smaller minimum size [e.g. 8]. Because there has also been a substantial time gap, 'Romer's Gap', between the latest Devonian tetrapods in the Famennian and the earliest known Carboniferous tetrapods in the Viséan, the differences have tended to be attributed to a mass extinction of tetrapods followed by radiation of a few surviving lineages [e.g. 41]. However, in recent years Romer's Gap has begun to close, with the discovery of Tournaisian tetrapod assemblages from Canada [42] and Britain [18,43,44] that reveal a diversity of tetrapods, some of them very small, living during the earliest Carboniferous. The presence of 'whatcheeriid-grade' tetrapods in the Famennian strongly suggests that the early Carboniferous tetrapod diversity has its roots in the Late Devonian, and this conclusion is further supported by the recent re-evaluation of the morphologically divergent limbless aïstopods (the earliest of which is the Viséan genus *Lethiscus*) as deep members of the tetrapod stem group [17]. It should also be noted in this context that the Tournaisian genus *Perittodus* consistently clusters with Devonian tetrapods in our phylogenetic analysis (figure 3) as well as those performed by Clack *et al.* [18].

In summary, *Brittagnathus* is not only the smallest tetrapod known from the Devonian, but also one of the most crownward. It adds to the growing body of evidence that the 'Carboniferous tetrapod radiation' had already begun during the Late Devonian, that tetrapods were not greatly impacted by the Hangenberg Event mass extinction, and that Romer's Gap may be nothing more than an artefact of poor sampling.

Data accessibility. The synchrotron microtomography dataset is available within the Dryad Digital Repository as a stack of TIFF images, divided into two parts on account of its size. The addresses of the datasets are https://dx.doi.org/10.5061/dryad.sqv9s4n0k and https://dx.doi.org/10.5061/dryad.vt4b8gtng [45].

The data for the phylogenetic analysis has been provided as electronic supplementary material in the form of three appendices: Appendix 1 is the character list, Appendix 2 is the data matrix in Word format and Appendix 3 is the data matrix in NEXUS format.

Competing interests. We declare we have no competing interests.

Funding. P.E.A. received funding for this work from a Wallenberg Scholarship awarded by the Knut & Alice Wallenberg Foundation.

Acknowledgements. We thank Sarah Finney (University of Cambridge) for mechanically preparing the specimen, Valéria Vaškaninová (Uppsala University) for generously making her ESRF beam time available for scanning the specimen, and Paul Tafforeau (ESRF) for assisting with the scan. Bent Erik Kramer Lindow (Natural History Museum of Denmark) provided the specimen number.

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
