## [Reviewer comments · Royal Society Open Science]

Review History

RSOS-192117.R0 (Original submission)

Review form: Reviewer 1 (Hans-Dieter Sues)

Is the manuscript scientifically sound in its present form?

Yes

Are the interpretations and conclusions justified by the results?

Yes

Is the language acceptable?

Yes

Do you have any ethical concerns with this paper?

No

Have you any concerns about statistical analyses in this paper?

No

Recommendation?

Accept with minor revision (please list in comments)

Comments to the Author(s)

Generally, this is an excellent paper on a phylogenetically really significant new taxon, and I can offer only two comments.

1. Please state that it is a right lower jaw/mandibular ramus in the text and in the figure legend.
2. The diagnosis is merely an abbreviated description and should be modified to be more informative. You should present a list of autapomorphies for *Brittagnathus* or a unique combination of character-states.

Review form: Reviewer 2 (Michel Laurin)**Is the manuscript scientifically sound in its present form?**

Yes

Are the interpretations and conclusions justified by the results?

Yes

Is the language acceptable?

Yes

Do you have any ethical concerns with this paper?

Yes

Have you any concerns about statistical analyses in this paper?

Yes

Recommendation?

Accept with minor revision (please list in comments)

Comments to the Author(s)

I enjoyed reading the paper, which generally reads well and is nicely illustrated (at least figures 1 and 2; I am less impressed with figure 3). It is a nice contribution to our growing knowledge of Devonian stegocephalians. I have profusely annotated the pdf file (so make sure that you get it), so I will concentrate here on the most important points.

My main comment is comments concern the phylogenetic analysis. First, the ordering scheme used (all characters unordered) is common but wrong. Cline characters should always be ordered. Simulations have shown that this increases resolution power and decreases the frequency of artefactual clades. Many authors think that not ordering is more careful and assumption-free, but it is not. Consider this: if you have mouse-sized, cat-sized, and elephant-sized taxa, is it more reasonable to think that to go from a mouse-size to an elephant-size, you have to go through a cat-size, or to think that you can jump from mouse-size to elephant-size directly? Only if you choose the latter is your assumption justified. But a glance at body size distribution in extant taxa plainly refutes this. For a detailed, simulation-based (and to a lesser extent, empirical) study about this, see Rineau et al. (2015, 2018). Among your characters, those that need ordering are 32, 50, 60 (probably), 65, 67, 73 (but states will need to be reordered first), 78, 79, 82, 95 (probably), 106, and 107.

Second, robustness analyses on the phylogenetic results (shown on Figure 3) are lacking but are really needed. After all, the main conclusions of this paper rest on these, and there are many equally-parsimonious trees yielding poorly-resolves strict consensus trees, so it is crucial to know just how robust the clustering of the new taxon with the Carboniferous stegocephalians is.

Fortunately, this is easy to do, at least for Bootstrap, which is implemented in PAUP. Bootstrap (Felsenstein, 1985) is the most important and reliable robustness index that can be obtained for parsimony analyses. If the authors wish to also carry out Bremer (decay) analysis (Bremer, 1988), this can be a nice complement, but it is technically a little more complicated, so I would not require it.

Finally, in the results, we need to know tree length (number of steps). This is the most basic, most important information (other than the topology) that we can get from a parsimony analysis and is required to make the study repeatable; after all, if others re-analyze the data and find different results, they need to know if they found shorter trees, or if they found longer, suboptimal trees.

I found a single mis-match between description and figures (2). The authors state "Much of its lateral side [of the adductor fossa] is flanked by the posterior end of the dentary.", but Figure 2 suggests that angular and surangular make up a far greater part of that wall than the dentary, which only contributes a narrow sliver on the anterior half of the dorsal edge of this wall.

This is not an anonymous review. If the authors have questions, they should not hesitate to contact me.

Best wishes,
Michel Laurin
michel.laurin@mnhn.fr

References

Bremer K. 1988. The limits of amino acid sequence data in angiosperm phylogenetic reconstruction. *Evolution* 42:795-803.

Felsenstein J. 1985. Confidence limits on phylogenies: an approach using the bootstrap. *Evolution* 39:783-791.

Rineau V., Grand A., Zaragüeta R., Laurin M. 2015. Experimental systematics: sensitivity of cladistic methods to polarization and character ordering schemes. *Contrib. Zool.* 84:129-148.

Rineau V., Zaragüeta I Bagils R., Laurin M. 2018. Impact of errors on cladistic inference: simulation-based comparison between parsimony and three-taxon analysis. *Contr. Zool.* 87:25-40.

Decision letter (RSOS-192117.R0)

06-Feb-2020

Dear Dr Ahlberg,

The editors assigned to your paper ("The smallest known Devonian tetrapod shows unexpectedly advanced features") have now received comments from reviewers. We would like you to revise your paper in accordance with the referee and Associate Editor suggestions which can be found below (not including confidential reports to the Editor). Please note this decision does not guarantee eventual acceptance.

Please submit a copy of your revised paper before 29-Feb-2020. Please note that the revision deadline will expire at 00.00am on this date. If we do not hear from you within this time then it will be assumed that the paper has been withdrawn. In exceptional circumstances, extensions may be possible if agreed with the Editorial Office in advance. We do not allow multiple rounds

of revision so we urge you to make every effort to fully address all of the comments at this stage. If deemed necessary by the Editors, your manuscript will be sent back to one or more of the original reviewers for assessment. If the original reviewers are not available, we may invite new reviewers.

- Data accessibility

<http://datadryad.org/submit?journalID=RSOS&manu=RSOS-192117>

- Competing interests

- Authors' contributions

- Acknowledgements

- Funding statement

Kind regards,

Andrew Dunn

on behalf of Dr Julia Brenda Desojo (Associate Editor) and Kevin Padian (Subject Editor)
openscience@royalsociety.org

Associate Editor's comments (Dr Julia Brenda Desojo):

Dear authors, this is an important and novel material as both reviewers suggested. However, a proper diagnosis should be provided (autapomorphies, compared characters, eg.), rather than an abbreviated description as the R1 suggested it. On the other hand, the phylogenetic section should be improve. Despite that you mention in pag 15 that the results presented should be considered as preliminary, a support analysis (such as Bootstrap, Bremer, etc.) should be done (see R2 suggestions).

Editor comments:

This is a beautifully written and illustrated piece on an intriguing jaw that seems definitely to represent a new animal. It has the deep jaw, sharp recurved fangs, and heterodont arrangement characteristic of basal tetrapods as well as basal amphibians and basal amniotes.

The reviewers have some concerns that we would like to allow you time to address. Please detail your responses to these. In addition, I have a few suggestions.

1. The term "advanced" in your title is not followed through in the manuscript. Could you be more precise about what the term means, or explain the point?

2. Figure 3 is reproduced at a very small size here and I wonder if you would like to put it in columnar format so it might be easier to read.

3. I agree with the reviewers that the diagnosis is a mix of things that may or may not be unique to this taxon; only unique features should be in the diagnosis. On the other hand it would be illuminating to detail the features that place your critter in the polytomy with the others.

Best wishes in revising.

Comments to Author:

Reviewers' Comments to Author:

Reviewer: 1

Comments to the Author(s)

Generally, this is an excellent paper on a phylogenetically really significant new taxon, and I can offer only two comments.

1. Please state that it is a right lower jaw/mandibular ramus in the text and in the figure legend.
2. The diagnosis is merely an abbreviated description and should be modified to be more informative. You should present a list of autapomorphies for *Brittagnathus* or a unique combination of character-states.

Reviewer: 2

Comments to the Author(s)

I enjoyed reading the paper, which generally reads well and is nicely illustrated (at least figures 1 and 2; I am less impressed with figure 3). It is a nice contribution to our growing knowledge of Devonian stegocephalians. I have profusely annotated the pdf file (so make sure that you get it), so I will concentrate here on the most important points.

My main comment is comments concern the phylogenetic analysis. First, the ordering scheme used (all characters unordered) is common but wrong. Cline characters should always be ordered. Simulations have shown that this increases resolution power and decreases the frequency of artefactual clades. Many authors think that not ordering is more careful and assumption-free, but it is not. Consider this: if you have mouse-sized, cat-sized, and elephant-sized taxa, is it more reasonable to think that to go from a mouse-size to an elephant-size, you have to go through a cat-size, or to think that you can jump from mouse-size to elephant-size directly? Only if you choose the latter is your assumption justified. But a glance at body size distribution in extant taxa plainly refutes this. For a detailed, simulation-based (and to a lesser extent, empirical) study about this, see Rineau et al. (2015, 2018). Among your characters, those that need ordering are 32, 50, 60 (probably), 65, 67, 73 (but states will need to be reordered first), 78, 79, 82, 95 (probably), 106, and 107.

Second, robustness analyses on the phylogenetic results (shown on Figure 3) are lacking but are really needed. After all, the main conclusions of this paper rest on these, and there are many equally-parsimonious trees yielding poorly-resolves strict consensus trees, so it is crucial to know just how robust the clustering of the new taxon with the Carboniferous stegocephalians is. Fortunately, this is easy to do, at least for Bootstrap, which is implemented in PAUP. Bootstrap (Felsenstein, 1985) is the most important and reliable robustness index that can be obtained for parsimony analyses. If the authors wish to also carry out Bremer (decay) analysis (Bremer, 1988), this can be a nice complement, but it is technically a little more complicated, so I would not require it.

Finally, in the results, we need to know tree length (number of steps). This is the most basic, most important information (other than the topology) that we can get from a parsimony analysis and is required to make the study repeatable; after all, if others re-analyze the data and find different results, they need to know if they found shorter trees, or if they found longer, suboptimal trees.

I found a single mis-match between description and figures (2). The authors state "Much of its lateral side [of the adductor fossa] is flanked by the posterior end of the dentary.", but Figure 2 suggests that angular and surangular make up a far greater part of that wall than the dentary, which only contributes a narrow sliver on the anterior half of the dorsal edge of this wall.

This is not an anonymous review. If the authors have questions, they should not hesitate to contact me.

Best wishes,
Michel Laurin
michel.laurin@mnhn.fr

References

Bremer K. 1988. The limits of amino acid sequence data in angiosperm phylogenetic reconstruction. *Evolution* 42:795-803.

Felsenstein J. 1985. Confidence limits on phylogenies: an approach using the bootstrap. *Evolution* 39:783-791.

Rineau V., Grand A., Zaragüeta R., Laurin M. 2015. Experimental systematics: sensitivity of cladistic methods to polarization and character ordering schemes. *Contrib. Zool.* 84:129-148.

Rineau V., Zaragüeta I Bagils R., Laurin M. 2018. Impact of errors on cladistic inference: simulation-based comparison between parsimony and three-taxon analysis. *Contr. Zool.* 87:25-40.

Author's Response to Decision Letter for (RSOS-192117.R0)

See Appendix A.

RSOS-192117.R1 (Revision)

Review form: Reviewer 1 (Hans-Dieter Sues)

Is the manuscript scientifically sound in its present form?

Yes

Are the interpretations and conclusions justified by the results?

Yes

Is the language acceptable?

Yes

Do you have any ethical concerns with this paper?

No

Have you any concerns about statistical analyses in this paper?

No

Recommendation?

Accept with minor revision (please list in comments)

Comments to the Author(s)

P. 4: It should be "Early Carboniferous" based on current stratigraphic usage because the Carboniferous is a period.

P. 7 - differential diagnosis: "Carboniferous tetrapods" is not informative as it represents a very broad range of clades. Please specify which taxa were used here for comparison.

Review form: Reviewer 2 (Michel Laurin)

Is the manuscript scientifically sound in its present form?

Yes

Are the interpretations and conclusions justified by the results?

Yes

Is the language acceptable?

Yes

Do you have any ethical concerns with this paper?

No

Have you any concerns about statistical analyses in this paper?

No

Recommendation?

Accept as is

Comments to the Author(s)

I think that the paper is ready now.

Decision letter (RSOS-192117.R1)

16-Mar-2020

Dear Dr Ahlberg:

On behalf of the Editors, I am pleased to inform you that your Manuscript RSOS-192117.R1 entitled "The smallest known Devonian tetrapod shows unexpectedly derived features" has been accepted for publication in Royal Society Open Science subject to minor revision in accordance with the referee suggestions. Please find the referees' comments at the end of this email.

The reviewers and Subject Editor have recommended publication, but also suggest some minor revisions to your manuscript. Therefore, I invite you to respond to the comments and revise your manuscript.

- Ethics statement

- Data accessibility

If you wish to submit your supporting data or code to Dryad (<http://datadryad.org/>), or modify your current submission to dryad, please use the following link:
<http://datadryad.org/submit?journalID=RSOS&manu=RSOS-192117.R1>

- Competing interests

- Authors' contributions

- Acknowledgements

- Funding statement

Because the schedule for publication is very tight, it is a condition of publication that you submit the revised version of your manuscript before 25-Mar-2020. Please note that the revision deadline will expire at 00.00am on this date. If you do not think you will be able to meet this date please let me know immediately.

To revise your manuscript, log into <https://mc.manuscriptcentral.com/rsos> and enter your Author Centre, where you will find your manuscript title listed under "Manuscripts with

Decisions". Under "Actions," click on "Create a Revision." You will be unable to make your revisions on the originally submitted version of the manuscript. Instead, revise your manuscript and upload a new version through your Author Centre.

on behalf of Dr Julia Brenda Desojo (Associate Editor) and Kevin Padian (Subject Editor)
openscience@royalsociety.org

Subject Editor Comments to Author:
Comments to the Author:

Per and Jenny, congratulations; we've accepted your revision with just the two minor corrections noted by the one reviewer. Thanks for submitting and stay well.

Reviewer comments to Author:

Reviewer: 1

Comments to the Author(s)

P. 4: It should be "Early Carboniferous" based on current stratigraphic usage because the Carboniferous is a period.

P. 7 - differential diagnosis: "Carboniferous tetrapods" is not informative as it represents a very broad range of clades. Please specify which taxa were used here for comparison.

Reviewer: 2

Comments to the Author(s)

I think that the paper is ready now.

Author's Response to Decision Letter for (RSOS-192117.R1)

See Appendix B.

Decision letter (RSOS-192117.R2)

19-Mar-2020

Dear Dr Ahlberg,

It is a pleasure to accept your manuscript entitled "The smallest known Devonian tetrapod shows unexpectedly derived features" in its current form for publication in Royal Society Open Science. The comments of the reviewer(s) who reviewed your manuscript are included at the foot of this letter.

on behalf of Dr Julia Brenda Desojo (Associate Editor) and Kevin Padian (Subject Editor)
openscience@royalsociety.org

Appendix A

The smallest known Devonian tetrapod shows unexpectedly derived features

Per E. Ahlberg & Jennifer A. Clack

Response to editors and referees

We thank the editors and referees for their helpful comments on our manuscript. Almost all of the suggestions have been implemented, as detailed below:

Associate editor:

We have improved the diagnosis and improved the phylogenetic analysis in accordance with the suggestions of Referee 2.

Editor:

1. What we are talking about is derived characters shared with post-Devonian tetrapods. We have replaced the word "advanced" with "derived".
2. The figure has been reformatted as you suggest.
3. The diagnosis has been modified as you and the referees suggest. *Brittagnathus* lacks autapomorphies but has a unique character combination.

Referee 1:

1. Done.
2. The diagnosis has been modified to make it more informative.

Referee 2 (Michel Laurin):

Comments on annotated manuscript:

Page 4, lines 2-3: Sorry about the mess-up with the matrix and character list, this has now been corrected! The “early Carboniferous forms that show plesiomorphic characters more typical of Late Devonian forms” are, specifically, the Tournaisian genus *Perittodus* [ref. 18] and the Viséan aïstopod *Lethiscus* [ref. 17]. These are now named in the text.

Page 5, lines 3-4 (and comments in report): Following your suggestion, we have re-run the analysis with characters 32, 50, 65, 67, 73, 82, 95, 106 and 107 ordered. 78 and 79 were not ordered because we did not feel that they formed unambiguous transformation sequences. Ordering the characters proved to be quite beneficial; it did not change the topology very much but reduced the number of trees to the point where the strict consensus tree revealed

phylogenetic structure that was previously only visible in the 50% majority rule tree. This applied to both smaller and the larger data set. We also carried out a bootstrap analysis of 1000 replicates on the small data set.

Page 5, line 7: This information is now provided.

Page 8, line 13: “mesially” has been changed to “medially”.

Page 10, lines 13-14: We expressed this clumsily: what we actually meant to say was that the dentary extends along most of the length of the adductor fossa, but of course if you think of the fossa as a three-dimensional object its lateral wall is indeed formed principally by the surangular and angular. The text has been corrected.

Page 10, line 23: Rephrased as “projects slightly above the dorsal jaw margin.”

Page 12, line 18-19: We are talking specifically about the differentiation of the adductor musculature here, so by definition the “crown tetrapod condition” must be that which is defined by the shared characteristics of extant amphibians and amniotes. However, if we assume that the changes in the shape of the adductor fossa that we see in the fossil record reflects the gradual differentiation of the adductor musculature (which seems reasonable), we can infer that this differentiation process started quite far down in the stem group: setting aside our disagreements about temnospondyls and anthracosaurs, the transformation is already under way in unambiguous stem-group members such as baphetids. We don’t think this point needs rephrasing.

Page 15: Because the phylogenetic analysis has been changed substantially, this whole section has been reworked, and has Figure 3 and its figure legend.

Page 16, lines 6-8: No, saying that all four are contemporary is an over-simplification. We have no direct evidence that either *Ichthyostega* or *Ymeria* is contemporary with *Brittagnathus*. This may seem like hair-splitting, but we feel that these details matter (or may come to matter in future) in the context of trying to understand the development East Greenland ecosystem. The East Greenland Famennian succession is thick and represents several million years, so a blanket claim that all these animals are contemporary could potentially be as inaccurate as saying that we are contemporaries of *Homo erectus*.

Page 19, line 2: “derived” changed to “crownward”.

Figure 2: “mesial” changed to “medial”.

Appendix B

UPPSALA
UNIVERSITET

Evolutionsbiologiskt centrum

Avd. för Evolutionär
organismbiologi

Evolutionary Biology Centre

Dept. of Evolutionary
Organismal Biology

Norbyvägen 18 A
SE-752 36 Uppsala
Sweden

From: Per Erik Ahlberg
Professor of Evolutionary Organismal Biology
Dept. of Organismal Biology
Uppsala University
Norbyvägen 18A
752 36 Uppsala
Sweden
Tel. +46 18 471 2641
Fax +46 18 471 6425
per.ahlberg@ebc.uu.se

Uppsala, 16 March 2020

Dear Anita, Julia and Kevin,

Jenny and I are delighted that you have seen fit to accept our manuscript RSOS-192117.R1 for publication and submit herewith our final revision. LSID registration numbers have been generated for the new genus and species, and the diagnosis has been modified in accordance with the suggestion of Referee 1. However, we have not changed “early Carboniferous” to “Early Carboniferous”, because apparently (according to Dave Millward, Jenny’s source for this) the Early Carboniferous is not an officially recognised unit anymore: it has been superseded by “Mississippian”.

Keep well and safe!

All the best,

Per Erik Ahlberg

Professor of Evolutionary Organismal Biology,
Uppsala University